# Morphology Control of Electrospun Brominated Butyl Rubber Microfibrous Membrane

**DOI:** 10.3390/polym15193909

**Published:** 2023-09-27

**Authors:** Tianxiao Zhu, Ruizhi Tian, Liang Wu, Dingyi Zhang, Leying Chen, Xianmei Zhang, Xiangyang Hao, Ping Hu

**Affiliations:** 1Engineering Research Center of Ministry of Education for Geological Carbon Storage and Low Carbon Utilization of Resources, Beijing Key Laboratory of Materials Utilization of Nonmetallic Minerals and Solid Wastes, National Laboratory of Mineral Materials, School of Material Sciences and Technology, China University of Geosciences, Beijing 100083, China; 1003200417@email.cugb.edu.cn (T.Z.); tianrz6632@gmail.com (R.T.); zhangdy97@163.com (D.Z.); chen.leying@outlook.com (L.C.); 2School of Electronics, Peking University, Beijing 100871, China; 2301112135@stu.pku.edu.cn; 3Key Laboratory of Arable Land Quality Monitoring and Evaluation, Ministry of Agriculture and Rural Affairs, Institute of Agricultural Resources and Regional Planning, Chinese Academy of Agricultural Sciences, Beijing 100081, China; zhangxianmei@caas.cn; 4Department of Chemical Engineering, Tsinghua University, Beijing 100084, China; hspinghu@tsinghu.edu.cn

**Keywords:** electrospinning, electrospun nanofibers, brominated butyl rubber, anti-bacterial property, conductivity

## Abstract

Brominated butyl rubber (BIIR) is a derivative of butyl rubber, with the advantage of high physical strength, good vibration damping performance, low permeability, aging resistance, weather resistance, etc. However, it is hard to avoid BIIR fiber sticking together due to serious swelling or merging, resulting in few studies on BIIR electrospinning. In this work, brominated butyl rubber membrane (mat) with BIIR microfiber has been prepared by electrospinning. The spinnability of elastomer BIIR has been explored. The factors influencing the morphology of BIIR microfiber membranes have been studied, including solvent, electrospinning parameters, concentration, and the rheological property of electrospinning solution. The optimal parameters for electrospinning BIIR have been obtained. A BIIR membrane with the ideal microfiber morphology has been obtained, which can be peeled from aluminum foil on a collector easily without being broken. Anti-bacterial property, the electrical conductivity of these membranes, and the mechanical properties of these samples were studied. The optimized BIIR electrospinning solution is Bingham fluid. The results of these experiments show that a BIIR membrane can be used in the field of medical prevention, wearable electronics, electronic skin, and in other fields that require antibacterial functional polymer materials.

## 1. Introduction

With the demand for emerging technologies such as medical materials and wearable electronic devices, butyl rubber has the advantages of high barrier properties, high heat resistance, and high damping properties [1,2,3]. At the same time, due to the introduction of bromine atoms, the reactivity of IIR is improved. This improved BIIR with self-adhesiveness, inter-adhesiveness, compatibility, and co-vulcanization has potential as a functional polymer material [4,5,6,7].

BIIR is a brominated product of butyl rubber. In addition to having the same good physical and chemical properties as butyl rubber, BIIR possesses more useful properties: (1) improved vulcanization rates due to the introduction of bromines, which increase the polarity of the molecular chain and (2) improved thermostability and compatibility with other rubbers [8,9]. BIIR is mainly used to make tyres, damping, and cork for medicine bottles. Due to the non-toxicity of BIIR, it has significant advantages in the field of medical applications. To answer the request from the BIIR manufacturer to develop extra values for BIIR, we produced BIIR for microfibrous membranes by electrospinning, which may be used in medical protective clothing, face masks, filtration, bandages, underwear, etc.

Electrospinning is one of the most common techniques to produce nanoscale or microscale fibers, which have excellent properties including a small diameter, high surface area, and high porosity [10,11,12]. Interest in electrospinning has been growing quickly because of the possibility to fabricate the products used in filtration, selective adsorption, etc. [13,14]. Electrospinning technology can continuously prepare nano/micro-sized fibers. Most of the current processing technologies cannot achieve this effect. Moreover, the fibers prepared using this method have the advantages of good continuity, ultra-fine diameter, and a high specific surface area, and this technology is simple, convenient, and cost-effective, so electrospinning technology has attracted increased attention. At present, there are hundreds of polymer materials, inorganic materials, and composite materials that can be used to prepare nanostructured materials through this method. In recent years, butyl rubber membranes have been fabricated by electrospinning [15,16].

However, there are still few studies on BIIR electrospinning. One of the important reasons is that BIIR is elastomeric, which is quite different from resin. It is hard to avoid BIIR fibers sticking together due to serious swelling or merging. In this work, electrospinning was used to prepare a BIIR microfibrous membrane with good morphology (without the presence of droplets or merged fibers). Our studies on the BIIR electrospinning technique were focused on factors which influence the morphology of BIIR fibers. These variables include the properties of BIIR, solvent, deposition distance, voltage, feeding rate of electrospinning, postprocessing techniques, etc. The properties of these BIIR microfibrous membranes were investigated. After selecting the optimal parameters for electrospinning BIIR, we studied BIIR as a highly functional material from the aspects of BIIR’s self-supporting properties, intrinsic antibacterial properties, electrical conductivity after incorporation of CNTs, and good mechanical properties.

## 2. Experimental

### 2.1. Materials

BIIR were provided by Cenway Polymer (Shanghai) Co., Ltd. (Shanghai, China). Tetrahydrofuran (THF), n-hexane and carbon tetrachloride (CCl_4_), atolein, sliced paraffin, and zinc stearate were purchased from Xilong Chemistry Co., Ltd. (Shantou, China). Carbon nanotubes (CNT) with a diameter of 12–15 nm and a length of 3–15 μm were purchased from Beijing Deke Daojin Technology Co., Ltd. (Beijing, China). Nano-silver and triclosan were purchased from Beijing Deke Island Gold (Beijing, China).

### 2.2. Characterization and Measurement

The morphology of the electrospun membrane was investigated by SEM (JEOL IT300, Akishima, Japan), with an accelerating voltage of 15 kV. The samples to be tested were pasted on the metal sample stage using conductive adhesive, and the surface was sprayed with gold. The rheological properties of BIIR electrospinning solvent were carried out by HAAKE™ RotoVisco™ 1 (ThermoFisher Scientific, Waltham, MA, USA). Antimicrobial performance tests were carried out in the Test Center of Antimicrobial Materials, Technical Institute of Physics and Chemistry, Chinese Academy of Sciences, according to Standard JIS Z 2801: 2012 [17]. The electrical performance of the fiber membrane with CNT peeled off from the aluminum foil was tested with a four-probe conductivity tester (KDA-1A, Kunde Technology, Kunming, China). The fiber membrane was cut into a sample with a diameter of 2 cm. The thickness was measured with a thickness gauge (SD-201, Jiurong Industrial, Shanghai, China). The mechanical performance of the fiber was tested with a highly sensitive force sensor. Tensile strength, elastic modulus, and elongation at break were tested on a computer-controlled electronic universal testing machine (CMT4304, Meister Industrial Systems, Eden Prairie, MN, USA), according to Standard GB/T 1040-2006/ISO 527:1993 [18].

### 2.3. Electrospinning

The electrospinning equipment (SS-2534) was purchased from Beijing Yongkang Leye Technology Development Co., Ltd. (Beijing, China).

BIIR mixtures including different auxiliaries such as atolein, sliced paraffin, zinc stearate, etc., were added to the solvent in a three-necked flask. Then, the mixtures were dissolved at 50 °C in an overhead stirrer at a speed of 400 rpm for three hours and then ultrasonic treated for 30 min to obtain homogeneous distribution.

The solutions were transferred to a 5 mL disposable syringe attached with a stainless-steel needle with an inner diameter of 0.51 mm. Electrospinning was conducted with a voltage from 10 to 20 kV. Distances from needle tip to collector (with aluminum foil on the surface) for the BIIR membrane were 15 cm, 20 cm, 25 cm, 30 cm, and 35 cm, respectively. Constant flow rates were 1.2 mL/h, 12 mL/h, 48 mL/h, and 96 mL/h, respectively.

After electrospinning, several methods were used to remove the residual solvent: heating, freezing, soaking in methyl alcohol, soaking in ethyl alcohol, lyophilization, and placing in a vacuum. There were many factors that influenced the morphology of the BIIR microfibrous membrane. In each group, only one parameter was adjusted, and others were kept fixed.

The local temperature was 19 °C and the relative humidity was 60%.

## 3. Results and Discussions

### 3.1. Property of BIIR

Table 1 shows the basic properties of three different BIIRs, and the SEM images of the electrospun fibers are shown in Figure 1. Due to the higher Mooney viscosity of BIIR-2501, its viscosity is higher than that of BIIR-2301 and BIIR-2302. It can be seen from Figure 1a that the fibers have begun to swell and stick together, which is due to the large bromine content of BIIR-2301 and its high viscosity, see Figure 1b,c. The obtained fibers are all smooth and have a uniform diameter. However, in the process of spinning, when using BIIR-2501, it is easy for it to get clogged at the needle. This is because the molecular weight of BIIR-2501 is too large, resulting in too strong a molecular entanglement ability, and it is difficult to form a jet under a certain voltage. Based on the above results, we selected BIIR-2302 for this study.

### 3.2. Rheological Property of BIIR Electrospinning Solution

A series of solutions with BIIR concentrations of 8%, 10%, 12%, etc. were prepared for electrospinning. Their rheological property was studied. Black curves are actual measurements, one for the shear ratio rising while the other for the shear ratio falling. The red line is their fitting line (Figure 2). Test results demonstrated that the shear stress obtained has good reproducibility and stability. Generally speaking, the BIIR electrospinning solution is Bingham fluid. The viscosity of 10 wt% electrospinning solution is 0.112 Pa·s (shear rate 100 s^−1^). The investigation may be helpful in the study of the spinnability of other electrospinning solutions, preliminary ascertainment of process conditions, and mechanization.

### 3.3. Solvent

In general, the viscosity and surface tension of a solution vary with the solvent. The viscosity of the high molecular polymer solution is mainly affected by two aspects: one is the molecular weight of the high molecular polymer and the other is the solution concentration. The surface tension of the solution is mainly affected by the type of solvent and the nature of the polymer. If the type of solvent used is not suitable, the surface tension will be too high, and even the electric field force cannot overcome the surface tension. This means that the polymer solution jet cannot be formed, and thus the micro-nano fiber cannot be formed. Therefore, the solution system with low surface tension is conducive to electrospinning, and it is important to find a suitable solvent [19].

According to the solubility of bromobutyl rubber, n-hexane, carbon tetrachloride, and tetrahydrofuran were considered, and the SEM images are shown in Figure 3.

When the solvent is hexane, the aluminum foil is almost entirely filled with beads and has almost no continuous fibers. This is because the surface tension of the solvent is 18.4 dyne/cm and the dielectric constant is 1.9 F/m. The viscosity of the solution is 122 mPa·s (the type of viscosity is dynamic, and the shear rate is 100 s^−1^, the same as below) and the stretching force of the electric field is much greater than the surface tension of the solvent. The jet is broken by the stretching force of the electric field almost as soon as it comes out of the needle, and the bead spraying phenomenon occurs. When carbon tetrachloride is used as the solvent, the surface tension of the solution is 35.2 dyne/cm, the dielectric constant is 2.2 F/m, and the viscosity of the solution is 202 mPa·s. The aluminum foil is covered with a string of bead structures, indicating that when the solvent is tetrachloride and carbon is used, the electric field force cannot overcome the surface tension to fully stretch the jet. When the solvent is tetrahydrofuran, the fiber surface is smooth and the diameter is uniform. The surface tension of the solution is 28.8 dyne/cm, the electric constant is 7.6 F/m, and the viscosity of the solution is 128 mPa·s. The electric field force overcomes the surface tension of the solution, the jet flows accordingly and becomes fully stretched. Therefore, we chose tetrahydrofuran as the solvent for this study.

### 3.4. Concentration of BIIR

In electrospinning, the solution concentration directly affects the solution viscosity, spinnability of the solution, and the morphology of the fiber. The viscosity of the solution is also greatly affected by the molecular weight of the polymer. High molecular weight polymers have a certain viscosity at low concentrations because the polymer chains with a high molecular weight are more prone to entanglement. Studies have shown that [20,21,22] as the molecular weight increases, the concentration suitable for electrospinning decreases continuously. When high molecular weight polymers are used for electrospinning, the usable concentration range must be narrow. Therefore, we fixed the parameters of electrospinning voltage, receiving distance, etc., and selected 8%, 10%, and 12% concentrations for electrospinning. The electron microscope image is shown in Figure 4.

The concentration of 10 wt% was better. Fibers failed to form at a concentration higher than 12 wt%. When at excess concentration, it was too sticky to form a Taylor cone even under high voltage. On the other hand, when the concentration was too low, continuous polymer fiber did not form (Figure 4a). Therefore, the processing window for BIIR electrospinning is narrow and the technology requirement is high. Therefore, a concentration of 10 wt% was selected for this study.

### 3.5. Electrospinning Process

In electrospinning, voltage is a very important control parameter. The electric field must be strong enough to overcome the surface tension of the droplet, which is the basic condition for the solution to be electrospun. When the electric field strength is sufficient, the electric field force on the droplet is enough to balance the surface tension, thereby forming a standard Taylor cone, leaving the needle and moving toward the receiving device, thereby forming a jet. The jet is stretched by the electric field and along with the expansion of the solvent volatile, gradually stretched into electrospun fibers with specifications at a nanometer scale. Only when the electric field strength reaches the lowest critical value of the spinnable polymer solution can the continuous fiber be obtained.

Keeping the parameters such as the concentration of the solution, the receiving distance, and the injection speed unchanged, the morphology of the electrospun fiber was investigated under different voltages such as 10 kV, 15 kV, and 20 kV, as shown in Figure 5. When the voltage is 10 kV, the rib-like structure can be clearly seen from the SEM image. This is because the voltage is too small and the electric field strength is too weak, causing the jet to not be fully stretched, and a large number of rib-like structures appear mixed in the fiber. When the voltage is increased to 15 kV, the obtained fiber is smoother and has a relatively uniform diameter. When the voltage increases to 20 kV, the fibers become messy and break. This is because the voltage at this time is much greater than the surface tension of the droplet at the needle, making the jet whip unstable, and the voltage intensity is much greater than the tensile strength of the fiber, causing the fiber to break. Therefore, when the concentration is 10%, the optimal voltage of electrospun bromobutyl rubber is around 15 kV.

Figure 6 exhibits SEM images of the electrospun BIIR membranes under different deposition distances. It is clear that with an increasing distance from tip to collector, the diameter of the fiber increased, and the porous structure vanished. BIIR fibers were relatively uniform. Since the optimized deposition distance ranged from 15 to 20 cm, to remove the residual solvent well and obtain drier fibers, we selected 20 cm for subsequent experiments.

Figure 7 shows the morphology of BIIR mats with various flow rates. A flow rate of 1.2 mL/h gave rise to an excellent morphology of BIIR microfibers. When the flow rate was too high, the solvent did not evaporate well. When the rate was too low, droplets of the solution did not form a Taylor cone and as a result no continuous fibers were formed.

### 3.6. Effect of Post-Treatment Process

To remove the excess solvent, the fiber membrane obtained in this study underwent a variety of treatment processes, such as heating, freezing, soaking in methanol, soaking in ethanol, freeze-drying, and vacuum placement. The processed electron microscope image is shown in Figure 8.

From Figure 8, when the fiber membrane is frozen (Figure 8b), the shape of the fiber is best preserved, while other solvent removal methods do not keep the original shape of the fiber membrane. Rubber has a high degree of resilience and viscosity, and the disappearance of fiber morphology and pore structure of the fiber membrane may be caused by the movement of the rubber molecules [23].

### 3.7. Release Agent

When using an ordinary approach, the breakage of the BIIR microfibers and the destruction of their porous structure frequently occur during electrospinning. These defects are caused by the high viscosity, resilience, and swelling of BIIR. Here, release agents such as atolein (Figure 9a), sliced paraffin (Figure 9b), and zinc stearate (Figure 9c), were added to the electrospinning solution to solve this headache defect, as shown in Figure 9. The effect of zinc stearate was obvious, and the average diameter of the electrospun BIIR microfibers with 8 wt% zinc stearate was approximately 3.42 μm with fewer nods and no bead formation. The microfiber morphology was significantly improved. With the proper amount of addition of zinc stearate, the swelling of BIIR was well depressed.

When the amount of zinc stearate is 3% (Figure 10a), rib-like structures appear on the fiber membrane and adhesion occurs between the fibers. When the amount of zinc stearate is 5% (Figure 10b), fewer ribbed structure are shown on the fiber membrane. When the amount of zinc stearate is 8% (Figure 10c), the fiber surface is smooth and uniform in diameter, and the porous structure is not damaged at all. The minimum amount of the release agent zinc stearate is 8%.

Because of the high cohesiveness of BIIR, the BIIR microfiber membrane sticks to the aluminum foil tightly. It is extremely difficult to peel off a BIIR microfiber membrane from aluminum foil without it being destroyed. To strip off the BIIR microfiber membrane from the aluminum foil, an alcohol aqueous solution of zinc stearate was sprayed on the aluminum foil and dried before electrospinning. The time for BIIR spinning was 1 h, 3 h, and 5 h, respectively. The BIIR microfiber membranes with different thicknesses on aluminum foil coated with stearic acid were successfully removed (Figure 11). As spinning time increased, the microfiber membrane became thicker and easier to remove. Then, the microfiber membranes were dipped in hot alcohol aqueous to remove residual zinc stearate and dried. At last, free-standing BIIR microfiber membranes were obtained. This lays a foundation for industrialized production.

After trying various process conditions, such as solvent, composition, concentration, electrospinning parameters (distance from tip to collector, flow rate, voltage, etc.), and postprocessing techniques (heating, freezing, soaking in low boiling point solvent, lyophilization, and placing in vacuum), we fabricated a BIIR membrane with an ideal microfiber morphology which is as good as resin, such as a representative electrospinning product made from PVA. Progress has been made compared with previous works [24,25,26].

### 3.8. Antimicrobial Property

Body-fitting materials such as wearable electronic functional materials require antibacterial properties. The strains used were *Escherichia coli* ATCC25922 and *Staphylococcus aureus* ATCC6538. After strain activation, membrane application, and elution, the culture conditions were 35 ± 1 °C, RH ≥ 90%, cultured for 24 ± 1 h, and the plate viable bacteria counting method was used. The test samples were six kinds of bromobutyl rubber spinning membranes, including a blank sample, nano-silver 1‰, nano-silver 3‰, triclosan 3‰, triclosan 5‰, and triclosan 7‰. The control sample was a standard PE sample, according to the standard regulations, and both the inspection sample and the control sample were cut into a size of 50 mm × 50 mm. The test results are shown in Table 2. Detailed study data have been discussed in our previous study [27]. According to the test results, we found that the BIIR membrane prepared by electrospinning without an antibacterial filler also has an antibacterial effect, and it has an intrinsic antibacterial effectiveness against *Escherichia coli* and *Staphylococcus aureus*. One reason is that the electron-deficient oxidized bromine atoms in BIIR have antibacterial activity. The other is that the BIIR membrane prepared by electrospinning has a large surface area, which makes it exhibit better intrinsic antibacterial properties.

Without antibacterial fillers, the electrospinning BIIR membrane shows intrinsic antibacterial effectiveness. This intrinsic antibacterial effect can reduce or even eliminate the addition of expensive antibacterial fillers such as nano-silver, effectively reducing costs while maintaining antibacterial performance and mechanical strength. BIIR microfibrous membranes can be used in bandages, underwear, medical dressings, protective suits, and surgical masks even for viruses, with no need to worry about a controversial nano-effect caused by texture filled with or coated with nanoparticles. In the previous COVID-19 epidemic, most of the cross-infection came from incomplete disinfection of the hospital environment, so antibacterial properties are of great significance for materials that fit the human body, such as protective clothing, gloves, etc. The BIIR membrane with good intrinsic antibacterial properties prepared in this study may have specific significance. Based on this significant intrinsic antibacterial performance, it may be used for the prevention of highly infectious viruses.

### 3.9. Electrical Properties

BIIR solutions with 1%, 5%, 8%, 9%, and 10% CNT were electrospun to produce BIIR membranes according to the best process condition as above. Their viscosity was 0.1137 Pa·s, 0.14818 Pa·s, 0.15719 Pa·s, 0.13205 Pa·s, and 0.11624 Pa·s, respectively. According to the conclusion of the second chapter, the polymer solutions with different contents of carbon nanotubes are all spinnable.

The scanning electron microscope images of the bromobutyl rubber fiber membranes with 1%, 5%, 8%, 9% and 10% added carbon nanotubes are shown in Figure 12. It can be seen from the figure that when 1%, 5%, 8%, 9%, and 10% carbon nanotubes are added, the morphology of the fiber remains good. Although there is a little tendon structure, there is no bead structure or hole structure. It also maintains a good performance and will not affect the breathability and moisture permeability of the fiber membrane.

The conductivity of the BIIR fiber membrane was tested. Only the fiber membrane with 10% CNT exhibits conductivity. In the process of testing the 9% to 10% CNT, the conductivity undergoes a sudden change (Figure 13). The samples with CNT less than 10% do not exhibit any conductivity. This is because when the addition of CNT is more than 10%, an integrated conductive network can be formed in the system. At this time, electrons are transported through conductive channels. When the addition of CNT is less than 10%, the integrated conductive network cannot be formed in the system. Substitute the measured data into the formula to calculate the conductivity of each group, and finally calculate the average value to obtain the conductivity of the fiber membrane as 11.2 μS·cm^−1^. The threshold value of percolation is 10% for the BIIR electrospun microfibrous membrane. Based on its intrinsic antibacterial properties, this allows BIIR to be used as a functional polymer material in sensors for wearable electronics after adding CNT.

When the amount of CNT added reaches a certain value, the conductive particles in the system will overlap with each other to form a three-dimensional conductive network loop [28]. The above is a macroscopic description of the conduction mechanism that forms a conductive loop, while electron transport is the microscopic process of the current generation. After the formation of the conductive network, the way the carriers move between the conductive particles determines the conductivity of the material. The electron transport process for conductive fillers is relatively complicated. Currently, there are a few main theories, which are the conductive channel theory, field emission theory, and tunnel effect theory [29,30]. The conductive channel theory is suitable for cases where the concentration of the conductive filler is high, and the conductive particles need to contact each other to realize the propagation of electrons. The field emission theory and the tunnel effect theory believe that this is achieved under the action of thermal vibration or an internal electric field. When the distance between conductive particles is less than 10 nm, electrons cross the potential barrier and propagate between the two conductive particles.

The peeling situation of the BIIR membrane with 10% CNT is shown in Figure 14. In order to explore whether the BIIR after adding 10% CNT can still be peeled off smoothly, we perform spinning and stripping according to the optimal parameters we obtained previously. We add 8% zinc stearate to the spinning solution as a release agent, and we conduct BIIR electrospinning of 10% CNT according to the optimal spinning parameters (electrospinning voltage is 15 kV, distance is 20 cm, solution concentration is 10 wt%, and flow rate is 1.2 mL/h). We used ethanol to dissolve the zinc stearate and applied the zinc stearate solution to the aluminum foil. When the ethanol evaporated completely, a thin layer of zinc stearate powder was left on the surface of the aluminum foil. After electrospinning, we soaked the peeled fiber membrane into the ethanol solution and washed away the zinc stearate. A complete BIIR fiber membrane with 10% CNT was obtained, as shown in Figure 14. The BIIR membrane with 10% CNT has a self-supporting structure without any rupture. This also lays the foundation for the subsequent commercial processes of functional polymer materials.

### 3.10. Mechanical Properties

The size of the sample is cut according to the standard requirements, with the gauge length of 25 mm ± 1 mm, the width of the narrow parallel part of 6 mm ± 0.4 mm, the thickness is measured by a digital thickness gauge, and the tensile speed is 20 mm/min. The main test contents are tensile strength, elastic modulus, and elongation at break.

As shown in Table 3, the average tensile strength of the blank group was 2.7 kPa, the elastic modulus was 430 kPa, and the elongation at break was 10.85%. Compared with the blank group, the tensile strength, elastic modulus, and elongation at break of the fiber were all improved after adding nano-silver, indicating that the interaction force between molecular chains was enhanced and the molecular chains were more tightly entangled. After the addition of triclosan, the elongation at break of the fiber membrane was significantly improved, but the addition of triclosan made the slip between the polymer chains easier, thereby reducing the elastic modulus. After adding 10% CNT, the elongation at break and tensile strength of the fiber membrane have been significantly improved. CNT is a good nano-reinforcement.

The mechanical properties of BIIR fiber membranes with nano-silver, triclosan, CNT, and without these fillers were measured. The BIIR with 10% CNT exhibits good tensile strength, elastic modulus, and elongation at break. It will not be easily damaged when used as a functional material. We studied the preparation process condition for BIIR microfibrous membrane electrospinning, mechanical properties, and antimicrobial properties of BIIR membranes, which lay a foundation for them to be used for wearable and self-powered sensors.

## 4. Conclusions

BIIR membranes with good microfiber morphology have been fabricated by electrospinning, which is as good as resin. Better technological conditions such as solvent, composition, concentration, electrospinning parameters, and postprocessing techniques, have been found. BIIR with a bromine content of 1.9% and Mooney viscosity of 32 and BIIR content of 10%, with solvent of THF, and release agent of zinc stearate of 10% are the best. When the solution is electrospun at 1.2 mL/h, 20 cm from the tip to the collector, the membrane has the best structure. The morphology of the membranes was the best after being frozen. After coating zinc stearate on aluminum foil, the membrane was easily peeled off. These results will help to produce BIIR microfiber or membranes on a large scale. The rheological properties of the BIIR electrospinning solution were tested. BIIR electrospinning solution is Bingham fluid, with a viscosity of 112 mPa·S. This provides the basis for research on the spinnability of other electrospinning solutions, preliminary ascertainment of process conditions, and mechanization. The BIIR microfibrous membrane shows significant intrinsic anti-bacterial properties, without any antibacterial filler or paint. With 10% CNT, it shows good electrical conductivity. This characteristic is also beneficial to conductive materials for wearable electronic devices. Due to its intrinsic antibacterial properties, it reduces or even does not require the addition of antibacterial fillers such as nano-silver. BIIR microfibrous membranes also exhibited high elongation at break after doping with nano-silver and CNT. This study provides insights to produce BIIR by electrospinning, which can be used in the field of medical materials, wearable electronic sensors, and electronic skin.

## Figures and Tables

**Figure 1 polymers-15-03909-f001:**
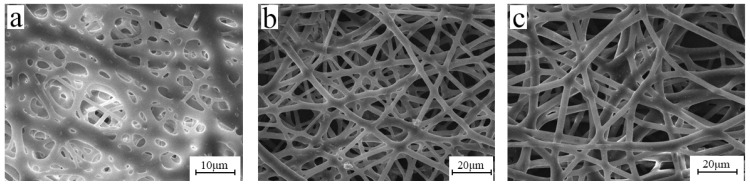
SEM images of electrospun fibers with different rubbers: (**a**) BIIR-2301, (**b**) BIIR-2302, and (**c**) BIIR-2501 (electrospinning voltage is 15 kV, distance is 20 cm, solution concentration is 10 wt%, and flow rate is 1.2 mL/h).

**Figure 2 polymers-15-03909-f002:**
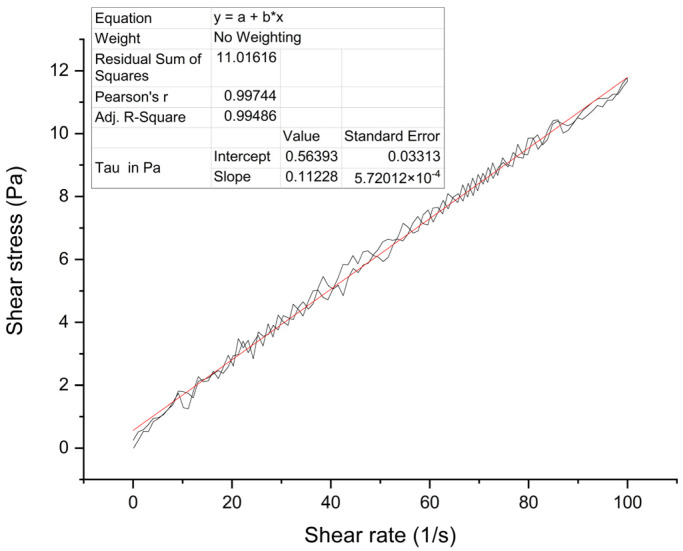
Rheological properties of 10 wt% BIIR electrospinning solution(The “*” sign represents the multiplication sign, and the red straight line is the fitting curve).

**Figure 3 polymers-15-03909-f003:**
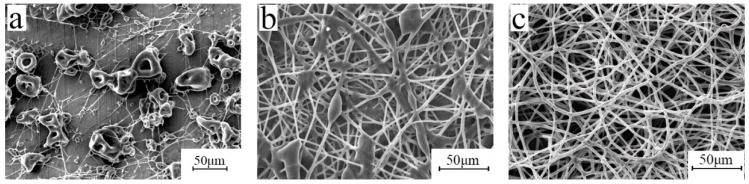
SEM images of bromobutyl rubber fiber membranes in different solvents: (**a**) n-hexane, (**b**) carbon tetrachloride, and (**c**) tetrahydrofuran (electrospinning voltage is 15 kV, distance is 20 cm, solution concentration is 10 wt%, and flow rate is 1.2 mL/h).

**Figure 4 polymers-15-03909-f004:**
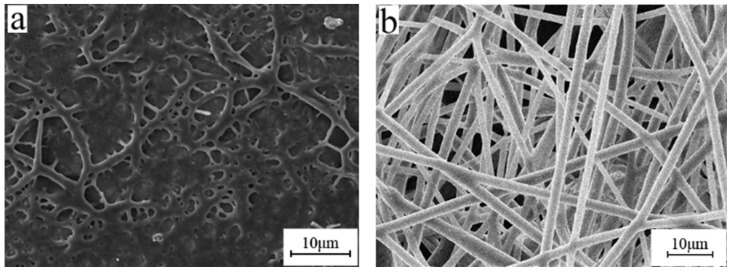
SEM images of electrospun BIIR microfiber with different concentrations (**a**) 8 wt% and (**b**) 10 wt% (electrospinning voltage is 15 kV, distance is 20 cm, and flow rate is 1.2 mL/h).

**Figure 5 polymers-15-03909-f005:**
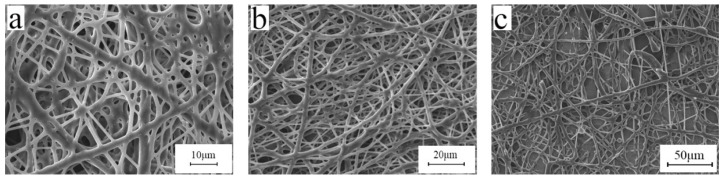
SEM images of bromobutyl rubber fiber membranes under different voltages (**a**) 10 kV, (**b**) 15 kV, and (**c**) 20 kV (distance is 20 cm, solution concentration is 10 wt%, and flow rate is 1.2 mL/h).

**Figure 6 polymers-15-03909-f006:**
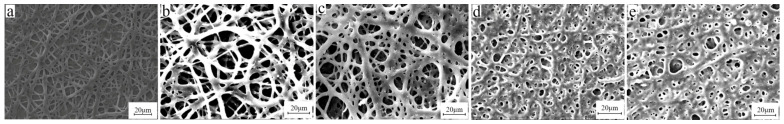
SEM images of electrospun BIIR membranes produced under different tip-to-collector distance of (**a**) 15 cm, (**b**) 20 cm, (**c**) 25 cm, (**d**) 30 cm, (**e**) 35 cm (electrospinning voltage is 15 kV, solution concentration is 10 wt%, flow rate is 1.2 mL/h).

**Figure 7 polymers-15-03909-f007:**
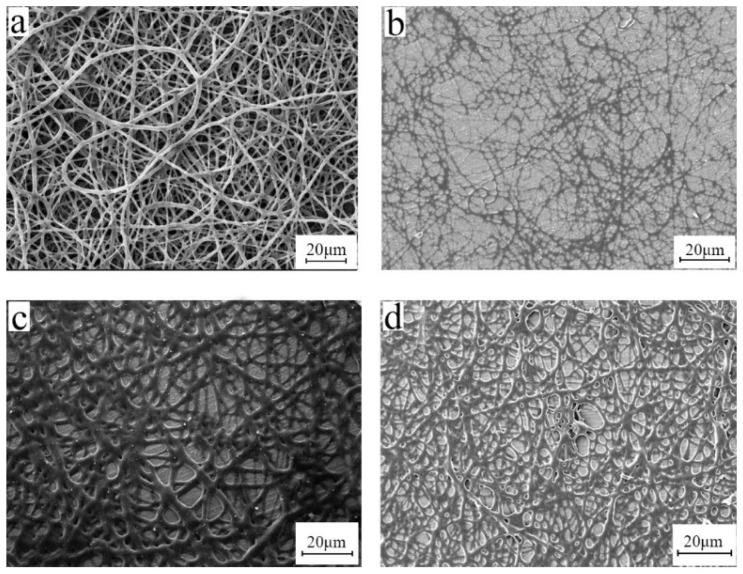
SEM images of electrospun BIIR membranes produced by different flow rates (**a**) 1.2 mL/h, (**b**) 12 mL/h, (**c**) 48 mL/h, and (**d**) 96 mL/h (electrospinning voltage is 15 kV, distance is 20 cm, and solution concentration is 10 wt%).

**Figure 8 polymers-15-03909-f008:**
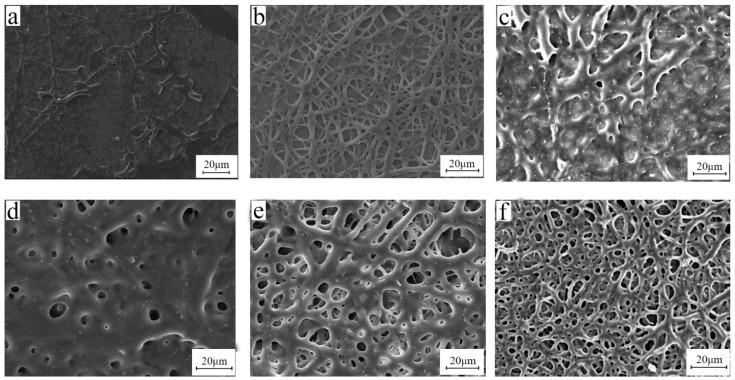
SEM images of bromobutyl fiber membranes in different post-treatment processes: (**a**) heating, (**b**) freezing, (**c**) soaking in methanol, (**d**) soaking in ethanol, (**e**) freeze-drying, and (**f**) place in vacuum (electrospinning voltage is 15 kV, distance is 20 cm, solution concentration is 10 wt%, and flow rate is 1.2 mL/h).

**Figure 9 polymers-15-03909-f009:**
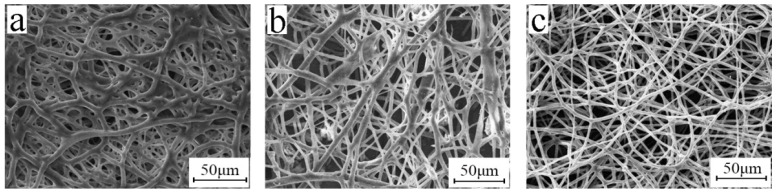
SEM micrographs of electrospun BIIR membranes with three types of release agents (**a**) atolein, (**b**) sliced paraffin, and (**c**) zinc stearate (electrospinning voltage is 15 kV, distance is 20 cm, solution concentration is 10 wt%, and flow rate is 1.2 mL/h).

**Figure 10 polymers-15-03909-f010:**
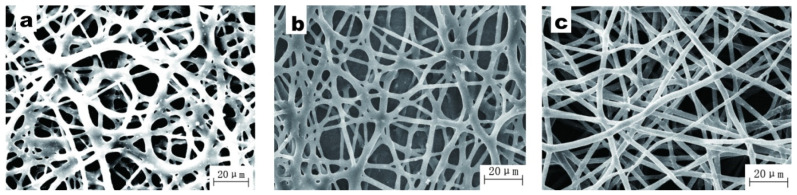
SEM micrographs of electrospun BIIR membranes with zinc stearate at (**a**) 3%, (**b**) 5%, and (**c**) 8% (electrospinning voltage is 15 kV, distance is 20 cm, solution concentration is 10 wt%, and flow rate is 1.2 mL/h).

**Figure 11 polymers-15-03909-f011:**
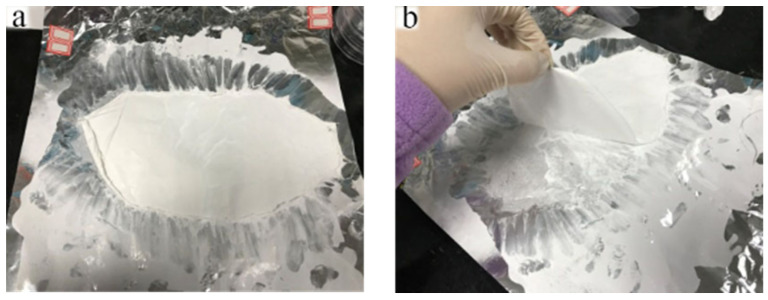
BIIR membrane stripped from aluminum foil. (**a**) BIIR membrane, (**b**) BIIR divestiture status.

**Figure 12 polymers-15-03909-f012:**
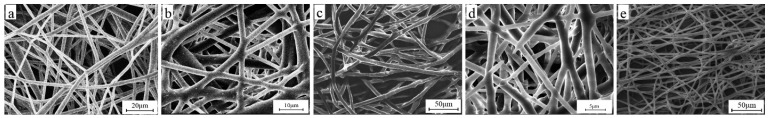
BIIR fiber membrane SEM with different carbon nanotube contents: (**a**) 1%, (**b**) 5%, (**c**) 8%, (**d**) 9%, and (**e**) 10% (electrospinning voltage is 15 kV, distance is 20 cm, BIIR solution concentration is 10 wt%, and flow rate is 1.2 mL/h).

**Figure 13 polymers-15-03909-f013:**
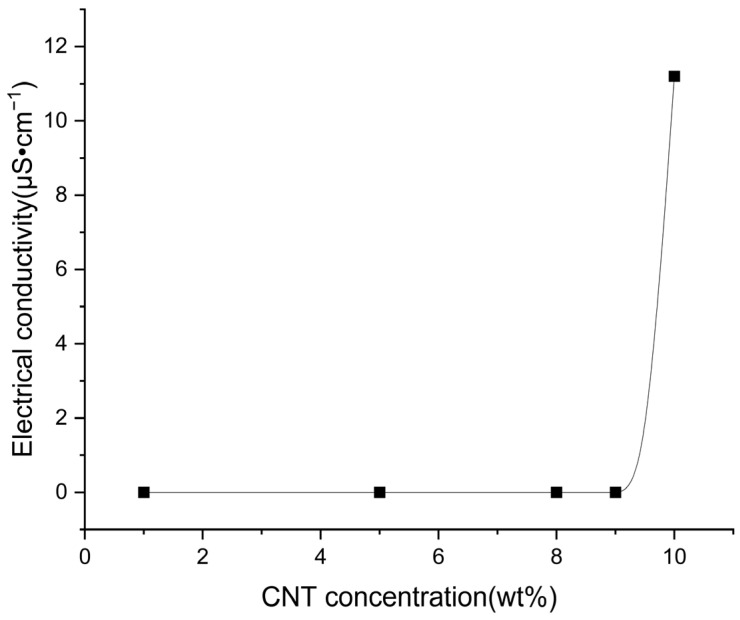
Electrical conductivity of CNT composites.

**Figure 14 polymers-15-03909-f014:**
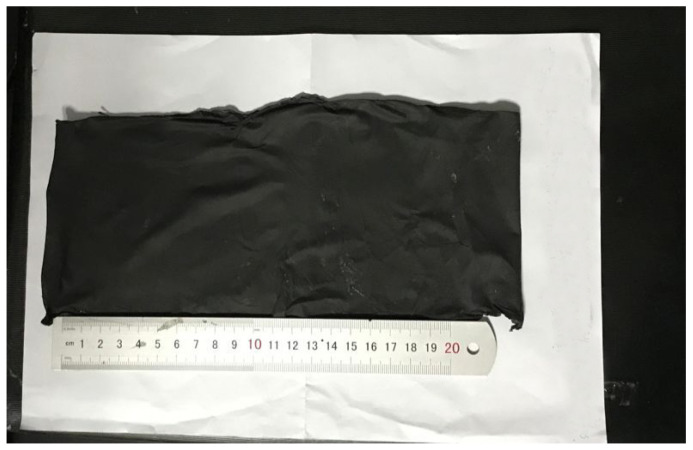
Peeled Bromobutyl Rubber Fiber Membrane with 10% CNT.

**Table 1 polymers-15-03909-t001:** Properties of BIIR.

BIIR No.	Bromine Content (wt%)	Mooney Viscosity (ML (1 + 8) of 125 °C) (MU)	Volatile Component (wt%)	Antioxygen Content (wt%)
2301	2.1 ± 0.2	32 ± 5	≤2 ± 5	≥2 ± 5
2302	1.9 ± 0.2	32 ± 5	≤2 ± 5	≥2 ± 5
2501	1.9 ± 0.2	46 ± 5	≤6 ± 5	≥6 ± 5

**Table 2 polymers-15-03909-t002:** Antibacterial performance test results.

	Antibactrial Rate (%)	*Escherichia coli*	*Staphylococcus aureus*
Sample			
Standard PE samples		--	--
blank		>99	>99
Nano silver 1‰		>99	>99
Nano silver 3‰		>99	>99
Triclosan 3‰		>99	>99
Triclosan 5‰		>99	>99
Triclosan 7‰		>99	>99

**Table 3 polymers-15-03909-t003:** Test results of mechanical properties of different BIIR fiber membranes.

	Tensile Strength (kPa)	Elastic Modulus (kPa)	Elongation at Break (%)
blank	2.7 ± 0.01	430 ± 2.15	10.85 ± 0.05
1‰ nano-silver	20 ± 0.10	570 ± 2.85	37.64 ± 0.19
3‰ nano-silver	11 ± 0.06	480 ± 2.40	33.00 ± 0.17
3‰ triclosan	10 ± 0.05	180 ± 0.90	56.72 ± 0.26
5‰ triclosan	18 ± 0.09	170 ± 0.85	70.99 ± 0.36
7‰ triclosan	34 ± 0.17	350 ± 1.75	63.31 ± 0.32
10% CNT	70 ± 0.35	270 ± 1.35	90.24 ± 0.45

## Data Availability

Data is contained within the article.

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
