# Peer review of "Morphology Control of Electrospun Brominated Butyl Rubber Microfibrous Membrane"

_polymers, 2023, doi:10.3390/polym15193909_

Round 1

Reviewer 1 Report

Polymers Manuscript Number:  polymers-2586620 Title: " Morphology Control of Electrospun Brominated Butyl Rubber Microfibrous Membrane" Author(s): Tianxiao Zhu, Ruizhi Tian, Liang Wu, Dingyi Zhang, Leying Chen, Xianmei Zhang, Xiangyang Hao *, Ping Hu

The paper is pretty well written and the scientific subject could be of interest from practical point of view, yet, the main concern of the current work is related to its sudden end. The recommendation is to revise the entire manuscript and the authors must explain more comprehensively their results.

Abstract:

The abstract is not a convincing one; It's too short. Also, does not quite draw attention of the reader. The authors need to provide an improved abstract of their work. The abstract should include the most important findings and highlight the innovativeness of this research.

Introduction:

The authors should clarify the research gap and the novelty of their work.

Results and discussion:

I suggest that 3.4. Concentration of BIIR to be brought before 3.3. Electrospinning process. I think that first we need to know the concentration and then vary the electrospinning parameters such as distance, voltage, etc. You discuss the SEM images in 3.3. Electrospinning process, but the reader does not know what concentration was used.

Page 5, line 161… Figure 4 is actually Figure 5, correct?

Page 6, line 170…. wouldn't it be better to use all the abbreviations?

I suggest that in each paragraph where SEM images are discussed, it should also be included in parentheses which image it is about. For example, page 6 line 175.”From Figure 6, when the fiber membrane is frozen (Figure 6(b)), or at page 7 line 191 “sliced paraffin (Figure 7(b), and zinc stearate (Figure 7(c))”; page 7 line 201 “When the amount of zinc stearate is 3 wt.% (Figure 8(a)), rib-like structure……”.etc.

Page 6, line 180, I think that a bibliographic reference should be added.

The rheological study should be presented before concentration. First we test the properties of the solutions and then we move on to testing the parameters of the electrospinning process.

3.8. Antimicrobial property

I don't see the point of this paragraph, especially since this study was presented in another work. Also, no figure, graph or table is given.

3.9. Electrical properties

A table listing the obtained data or at least a graph should be given.

The discussions must describe more accurate the physical processes which take place.

Please provide the error bar (Table 2).    

Pay attention to the quality of the images.

Author Response

Thank you for your review, response letter is attached below.

Reviewer 2 Report

Dear Authors,

Thank you very much for submitting the article entitled "Morphology Control of Electrospun Brominated Butyl Rubber Microfibrous Membrane".

The study seems to be interesting and informative, but needs to be corrected due to the lack of several important questions. In the current view the study is fragmented, incompleted, and could not be recommended for publication.

Please, find my comments below.

1. P. 2, subsection 2.2.

This section should be written more detailed, e.g. applied voltage and working distance of the SEM, the sample coating, the quantity of samples and measurements for each method.

2. P. 2, subsection 2.3.

Please, demonstrate the concentration of BIIR and the solvent/solvents which were used.

Moreover, the electrospinning system should be described (industrial or handmade?).

At the same time, temperature and relative humidity must be provided.

3. P. 3, subsection 3.1.

The Authors discussed the viscosity and surface tension, but did not provide any experimental data. Moreover, electrical conductivity of the spinning solutions must be measured.

Please, demonstrate the measured parameters, e.g., in the Table. Note, that SD should be provided.

4. P. 3, Figure 1.

The electrospinning parameters of each sample should be provided.

5. P. 4, subsection 3.3.

Please, provide the electrospinning parameters more carefully: in the first paragraph the applied voltage should be provided, in the second paragraph - the distance.

6. P. 4, Figure 3

The quality of the figure must be improved. Moreover, I kindly recommend the Authors to add min and max diameter into each histogram. Moreover, SD must be demonstrated.

7. P. 5, line 145

The Authors wrote "The morphology of BIIR mats made at 10, 15, and 20 kV were studied, and 15 kV was the best". The table with the morphological parameters must be provided. How do you evalue "the best" fibers? Did you use any additional parameters, such as polydispersity index? Each statement must be proved.

Moreover, I kindly recommend including the diagram (or plot, table) which demonstrates the influence of the electrospinning parameters on the process and on the fiber properties.

8. P. 5, line 161

"The electron microscope image is shown in Figure 4". In my opinion, Figure 5 is the more correct variant for this reference.

9. P. 6, Figure 6.

Please, demonstrate the electrospinning parameters and the rubber concentration of the initial samples.

10. P. 6, subsection 3.6.

Please, highlight the concentration of rubber into spinning solution and the electrospinning parameters. At the same time, the changes detected during the electrospinning with additive should be discussed. Moreover, it is important to demonstrate the differences of the key parameters (viscosity, electrical conductivity, surface tension) between initial solutions and loaded solutions. Note, that key parameters should be demonstrated in each case, especially at the initial stage of study.

The morphological parameters with the SD should be provided in comparison with blank (unloaded) fibers.

11. P. 8-9, subsection 3.8. Antimicrobial property

The figures or diagrams must be provided.

12. P. 10, table 2

SD must be provided.

13. P. 12, 4. Patents

This section now is blank and should be omitted.

The text should be carefully checked and improved.

Author Response

(The authors gave the same response as above.)

Round 2

Reviewer 2 Report

Dear Authors,

Thank you very much for the submission of the revised manuscript and for your detailed answers.

The study was improved and corrected. At the same time I have additional suggestions (please, see below).

1. P. 5, line 161, 165, 170

Please, use the dot (·) symbol for the viscosity units (mPa·s)

Also, please, specify the type of viscosity: dynamic, intrinsic, relative, specific, reduced, kinematic, absolute. In the first case, please, add the shear rate.

2. P. 11, Figure 13

It is more preferable to use the term "Electrical conductivity" instead of Electronic conductivity. Please, correct the caption and the figure.

Moreover, please, use μS to remove 10-5 and 10-6.

I recommend this study for publication after minor revisions.

Author Response

The response letter is below. Thank you for review.
